# Prevention of Musculoskeletal Injuries in Gastrointestinal Endoscopists

Tadej Durič [1,*] , Ivana Cibulkova [2] and Jan Hajer [2]

1 3rd Medical Faculty, Charles University, Prague 100 00, Czech Republic
2 Department of Internal Medicine, Kralovske Vinohrady University Hospital, Šrobarova 1150/50, Prague 100 00, Czech Republic; ivana.cibulkova@fnkv.cz (I.C.); jan.hajer@fnkv.cz (J.H.)
* Correspondence: tadej.duric@gmail.com; Tel.: +420-38640394309

**Abstract:** Gastroenterologists are exposed daily to musculoskeletal (MSK) stress during upper and lower gastrointestinal endoscopy, both during routine endoscopies and during long, demanding therapeutic procedures. There is evidence that endoscopy-related MSK injuries are becoming more common, particularly in the back, neck, shoulders, elbows, and hands. The aims of this study were to identify the most stressed muscle groups during endoscopy; to measure their activity using surface electromyographical (EMG) sensors; to detect areas of muscle overload; and to identify the number of microbreaks taken in specific muscles. Furthermore, we measured differences in the loading of these muscle groups with and without the use of special support systems such as a belt-like holder. Measurements were performed on 15 subjects (7 experienced endoscopists and 8 non-endoscopists). Due to this small sample size, inside each group, we achieved inconclusive results regarding statistically significant differences in different muscle groups. We increased the sample size by comparing all participants with and without the belt support system, disregarding their endoscopic background. There was a statistically significant difference ($p < 0.05$) in muscle tension and in levels of microbreaks in the muscles of the left forearm, biceps, and trapezius muscles. No statistically significant difference was observed in the muscle tension and level of microbreaks in the left deltoid muscle ($p > 0.05$). We hypothesize that the increased level of muscle loading and decreased level of microbreaks in the deltoid muscle are due to different muscle activity and different shoulder movements. Additionally, the deltoid muscle is not connected to the kinetic chain of body posture and stabilization. It is our belief that MSK injuries in gastrointestinal (GI) endoscopy can be prevented with the use of a belt-like support system.

**Keywords:** GI endoscopy; musculoskeletal injuries; prevention; support belt holder; surface EMG

## 1. Introduction

Gastrointestinal (GI) endoscopy forms an important part of the daily practice of a gastroenterologist. Its use and applicability have shifted to be therapeutic as well as diagnostic, and it makes up a significant proportion of the workload of the GI practitioner. The basic design of a flexible endoscope has remained practically the same for the last couple of decades [1], and the complexity of modern endoscopic procedures has increased. Musculoskeletal injuries amongst endoscopists are therefore on the rise. According to a systematic review of 13 studies by Young et al. [2], 39–89% of surveyed endoscopists reported pain and/or injuries related to endoscopy. Common areas of pain were the back (15–57%), neck (9–46%), shoulders (9–19%), elbows (8–15%), and hands/fingers (14–82%). The risk factors included the procedure volume, time spent performing endoscopies, cumulative time in practice, and endoscopist's age. The experimental studies showed that forces and loads placed on endoscopists' bodies during procedures place them at risk of occupational injury. The areas of pain differed between novice and experienced endoscopists, implying different mechanisms of injury. Villa et al. [3] described the importance of proper endoscopy training

in young fellows. Of the two groups in their study, the one without formal ergonomic training had a significantly higher percentage of musculoskeletal (MSK) injuries related to endoscopy. Against the background of this emerging and growing problem, basic recommendations were published by Amandeep K. Shergill [4]. They address basic ergonomic principles related to endoscopy and propose organizational and spatial solutions with the intent of minimizing the possibility of occupational injury. Similar publications have emerged subsequently to address the problem and offer similar solutions [5–10]. A solution for better working conditions during everyday endoscopic procedures has been sought. When performing endoscopy, the upper extremities and body core (posture) perform the most activity. Muscle activity can be considered either dynamic or static. When exercising dynamic activity, changes in body position (e.g., when walking) allow sufficient blood and lymphatic flow. In contrast, static activity decreases blood and lymphatic flow, resulting in pain and discomfort [11]. Static muscle loading is when a muscle contracts but does not result in movement of that part of the body. It can occur during many work activities, such as when carrying an object, and is the main muscular activity used when performing endoscopic procedures. An endoscopist usually has a static stance, holds the endoscope in the left hand, and maneuvers the shaft with the right hand. No major body movements are made by the endoscopist. Muscles therefore experience a load of force, resulting in a gradual switch to anaerobic metabolism, hypoxia, and muscle overload [12]. To prevent muscle overload, microbreaks of the muscle play a crucial part. Microbreaks are 1–2 s interruptions in muscle tension every 60 s. During use, muscles contract around the blood vessels, inhibiting blood flow. If tension is maintained without interruption (static effort), blood as well as lymph flow is continuously interrupted at a time when increased flow is required [11]. Microbreaks are scheduled rest breaks taken to prevent the onset or progression of cumulative trauma disorders. It has been reported that microbreaks reduce muscle discomfort, particularly when breaks are taken at 20 min intervals [13].

The objectives of our study were as follows: to describe the specific muscles involved in performing endoscopic procedures; to measure their activity with surface electromyography (EMG) sensors during endoscopic procedures; to detect areas of muscle overload; and to explore the number of microbreaks taken in specific muscles, and suggest solutions to prevent muscle overloading (and subsequent MSK injuries).

## 2. Materials and Methods

### 2.1. EMG Measurements of Basic Endoscopic Movements

Our first goal was to describe the specific muscle activity during basic endoscopic movements. With the collaboration of a physiotherapist, we analyzed body posture and movements during endoscopic procedures. We identified the muscle groups that are involved in practicing endoscopy. As outlined before, the endoscopist has a mainly static stance, where they hold the head of the endoscope and its controls with the left hand, and maneuver the shaft with the right hand. More static muscular activity occurs in the left arm and body posture. Our focus therefore shifted to these areas. We targeted muscles that are responsible for movement of the left arm, which are also connected to the vertebrae and therefore affect body posture. Surface EMG sensors were applied to the forearm extensors, forearm flexors, biceps, triceps, and deltoid muscles. We used skin surface EMG detectors from Biometrics Ltd. (Newport, UK) and applied them to the skin using elastic fixation nets. Data were processed using the Biometrics DataLite EMG System version 10 (Biometrics Ltd.). This software offers the synchronization of EMG measurements and video recordings. A camera was placed in front of the participant and recording commenced with the EMG measurements. Each change in EMG potential could be explained by identifying the specific movement on video. The accuracy of the measurements and their interpretation allowed for high-quality data gathering.

A multifactorial approach was required to design the measurement protocol. Firstly, the team responsible for taking measurements prepared the environment for the participants and set up all necessary equipment. To establish synchronized EMG and video measure-

ments, the sensors, the camera, plug-in interfaces, and a computer were required. After correctly establishing all of the measurement parameters, we explained the basic endoscopic maneuvers to each participant. Each participant performed basic maneuvering with the endoscope for 1 min. We recorded EMG measurements from an endoscopist and a non-endoscopist. The non-endoscopist participant received formal training on basic endoscope handling and movement. Preparation for each measurement was time-consuming due to all of the elements in the measurement chain and the process of synchronizing them (room preparation, application of EMG sensors, camera and software initiation and synchronization). We recorded data from two participants, who also repeated basic movements multiple times. Data were presented as a combination of EMG curves (Figure 1) and a video file.

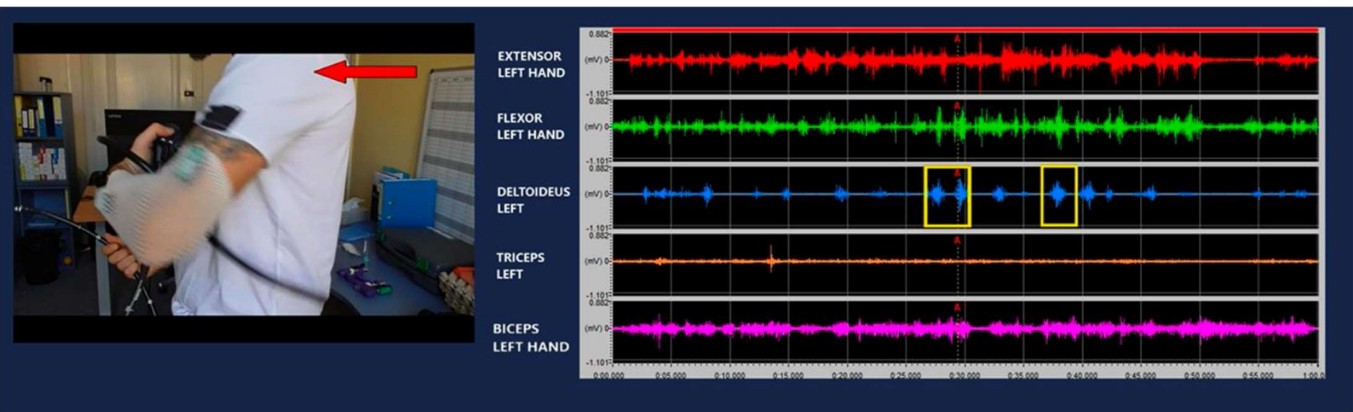

**Figure 1.** The arrow points to the deltoid muscle; the yellow brackets outline areas of increased muscular activity of the deltoid muscle when moving the scope in a circular shape.

## 2.2. Development of Standardized Procedure

### 2.2.1. Training Box Development and Manufacturing

We standardized the movements involved in this investigation to equalize the conditions for further participants. We aimed to recreate an environment that emulates basic endoscopic movements and activities; has a simplistic design; is portable; and is user-friendly and environmentally friendly. A previous article described a special course box that was designed to assess the endoscopic abilities of an individual. Participants had to perform five different tasks which included basic endoscopic movements and actions (snare manipulation, pinpoint accuracy, loop solving, moving objects, and forceps manipulation). Points were awarded and time was measured for the completion of each task [14]. However, this assessment tool had a complicated design with many entry points and different mechanics. Our idea was to have a training box with one entry point and the potential to perform different tasks. We aimed for it to be light, portable, and durable. Through trial and error, we designed a special training box with a single entry point for the endoscope. The box was a cube-like structure, with a side entry point and four doors fitted on different sides. A structure of tubes was designed through the entry point. These tubes allowed the endoscope to reach 3 different positions/doors. On the inside of each door, a plate with pins and holes was fitted. Through different positions of pins, different tasks could be performed to imitate basic maneuvering of the endoscope in everyday practice. Repeated experiments were performed to adjust the tubes and plates in the correct position to allow for task completion. The box was made primarily out of wood, with additional plastic parts fitted inside the box. The plates used for the tasks were manufactured with a 3D printer. The training box was painted black on the inside, with the aim to emulate real-life conditions when performing an endoscopy. The endoscope was therefore the only source of white light inside the training box (Figures 2–6).

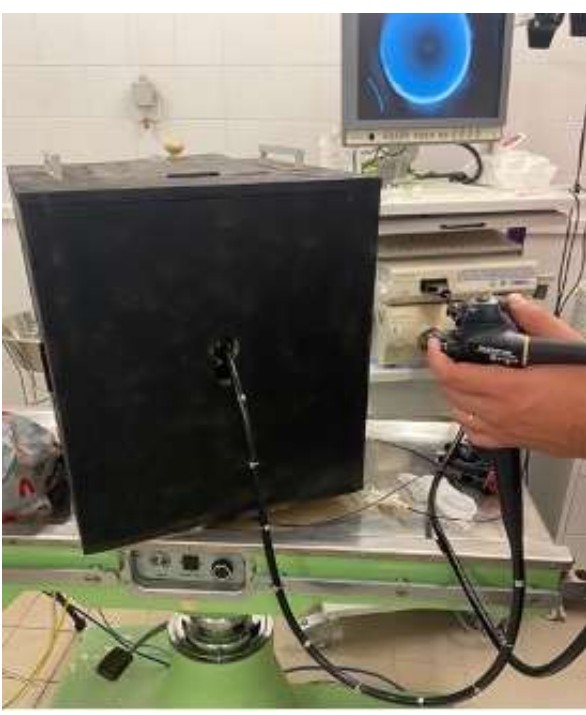

**Figure 2.** Training box frontal view with inserted endoscope.

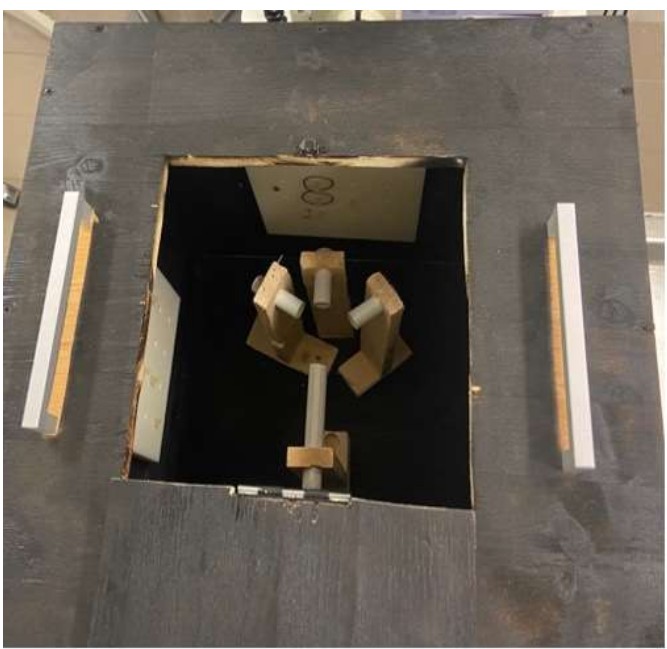

**Figure 3.** Training box ground view with a system of tubes for navigation to each task.

### 2.2.2. Task Design

The aim of the endoscopic training box was to act as a training model to simulate everyday endoscopic maneuvering and tasks. Diagnostic endoscopy, which represents the bulk of endoscopic procedures, requires good maneuvering skills and good postural stability. Therapeutic endoscopy also requires good maneuvering skills and postural stability, with the addition of precise movements and a lengthier procedure time. We designed a box which simulates tasks from diagnostic and therapeutic procedures: reaching certain positions; moving objects; and performing various tasks. The tasks were designed to emulate the basic principles of endoscopic maneuvering and tasks performed in everyday

practice. The tasks would differ in difficulty and nature. As in real life, the aim was to emulate an endoscopic procedure: the introduction of the endoscope, its insertion, its maneuvering, reaching a certain target, and biopsy completion. The first task was to reach all three task stations. The second task was drawing the number eight between two pins. The third task involved moving an object (a rubber band) with forceps from one pin to an adjacent one. Second-person assistance was provided to each participant. After each task was completed, the participant had to withdraw the endoscope to the starting position in the introduction tube and proceed with the next task. By creating the training box and its tasks, we standardized the movements for each participant (Figure 7).

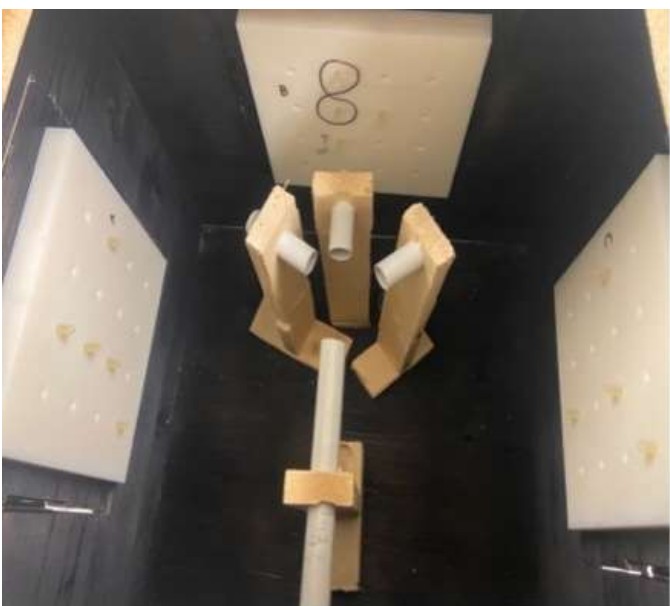

**Figure 4.** Ground view inside the training box. Visible is the set of tubes that lead to different platforms with different tasks.

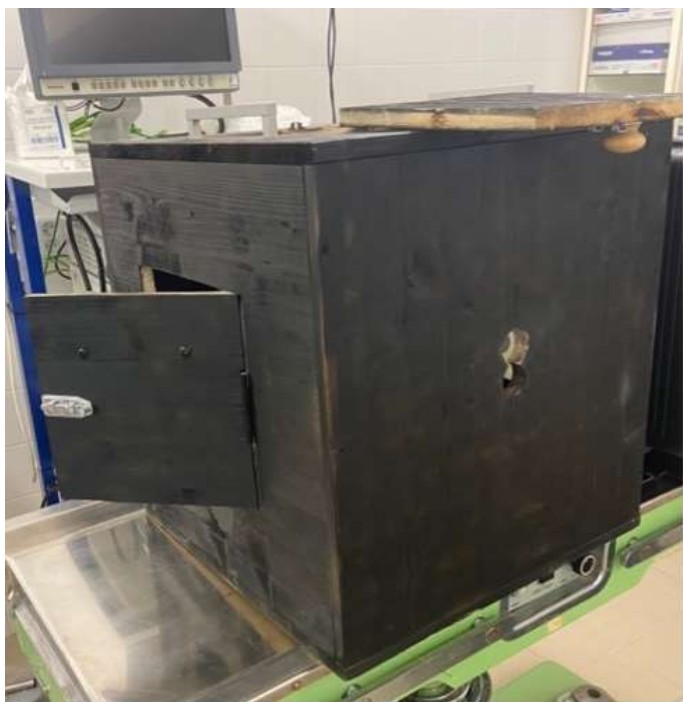

**Figure 5.** Training box from the lateral view with the side door opened.

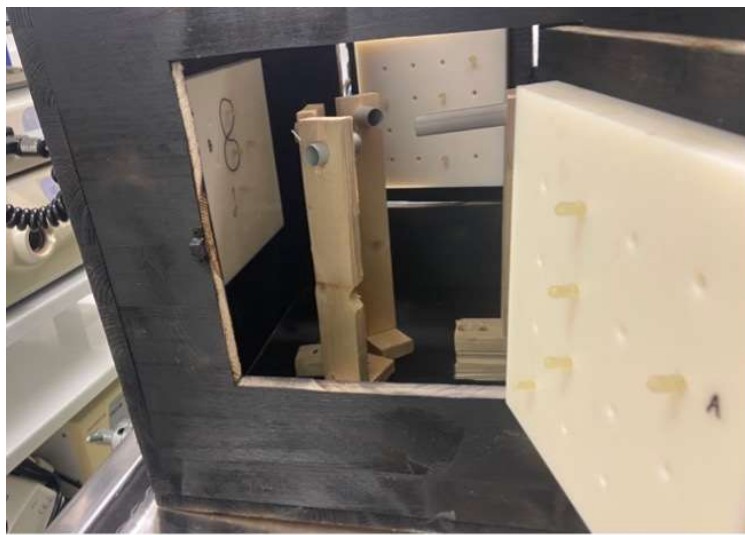

**Figure 6.** Training box from the lateral view through the side door. On the door is a fitted platform with pins used to exercise movements. Through the door, a set of tubes and other platforms with different tasks are visible.

### 2.3. Muscle Tension and Microbreak Measurements

To measure muscle loading, muscle tension, and the number of microbreaks, we initially considered the use of skin surface EMG sensors. They have specific limitations, however, namely the time spent to prepare and mount all of the sensors; the fact that measurements are taken only at one point of skin contact; and the instability of the attached sensors. We therefore decided to use a specially designed suit with integrated surface EMG sensors (Figure 8). The suit was constructed from elastic synthetic material, with an inner layer covered with a strip surface. The EMG sensors were connected to Bluetooth plugins. The plugins were attached to the suit externally and then connected to the software.

The Ergosuit enabled data to be gathered simultaneously from multiple muscles and muscle groups, giving insight into overall muscle performance. EMG curves and video recordings were synchronized with the provided software. The Ergosuit, with the accompanying software, Ergolink and Ergoanalysis version 1, are products of Myontec Ltd. (Kuopio, Finland). Their smart clothes (Ergosuit) record upper body muscle activation (EMG), motion, and heart rate, which can be combined with video recordings for easy and direct ergonomic assessment (Figure 9). Analysis of the load distribution, static load, microbreaks, and overhead and bending positions were performed with the algorithms provided.

The measurement protocol changed with the use of the suit. Each participant had to undress their upper body, to ensure close contact with the suit's sensors. The required level of privacy was provided. Before dressing, the suit was disinfected and scrubbed inside. Additionally, just before application, the sensors were wiped with a damp cloth soaked in sterilized water, to ensure better skin contact with the EMG sensors. After dressing, a set of plugins were attached to the suit. Each plugin was connected via Bluetooth to the software interface and synchronized with the other suit plugins. Additionally, a phone camera was set up to record the whole procedure for each participant. At the beginning of the measurements, video synchronization was performed for each participant. Compared to the previous skin surface EMG sensors, this method was also time-consuming. However, it provided a higher amount of data and greater sensor stability.

As outlined previously, static muscle load and the number of microbreaks play an important role in discomfort and injury prevention. The level of muscle loading and percentage of microbreaks while performing standardized tasks were measured in each participant. After initial analysis, results for each participant were provided in a combination graphic and numerical report. A portfolio of each participant was formed, with the measured areas depicted as colored numerical values in Figure 10.

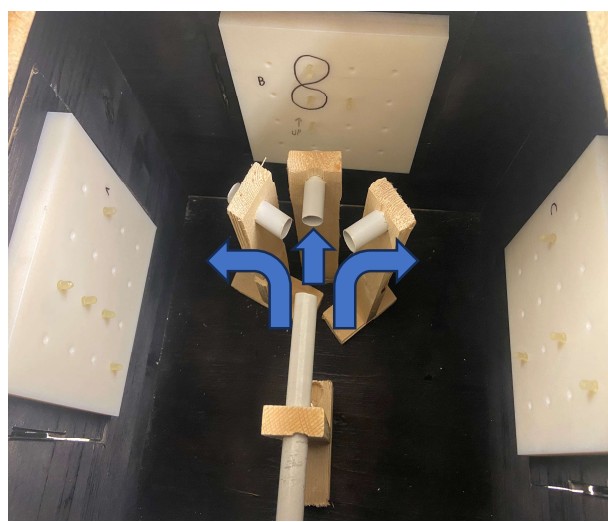

Task 1: To reach all 3 positions (A, B, C).

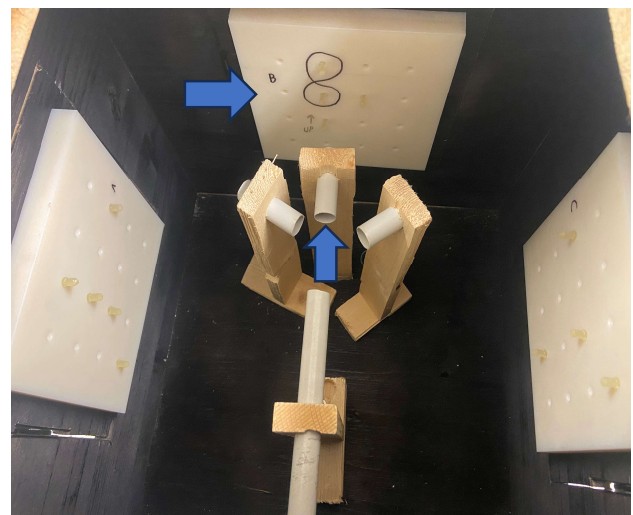

Task 2: To draw the number 8 with the tip of the endoscope. For easier navigation, number 8 was drawn around 2 pins.

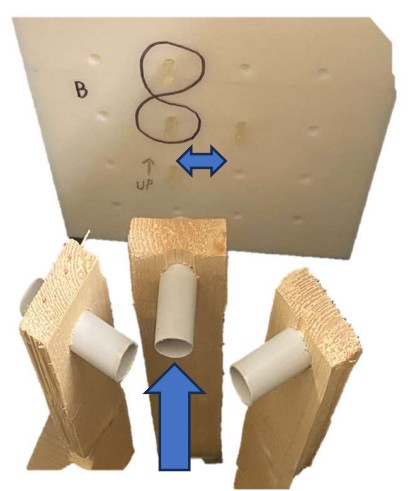

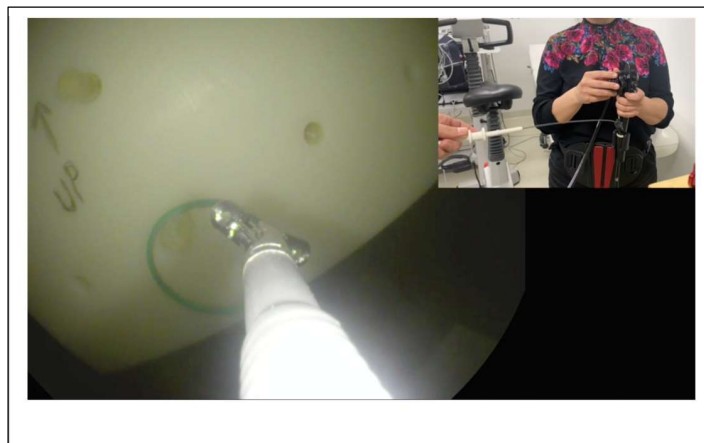

Task 3: To move an object (rubber band) from one pin to the next using biopsy forceps.

**Figure 7.** Task design.

The first part of Figure 10 shows measurements of muscle loading during the standardized tasks. Muscle load was provided as a percentage of maximum voluntary contraction of the muscle. As a reference, we used data from a company which specializes in ergonomic assessment and analysis (Premedis s.r.o., Liberec, Czech Republic). In the years between 2004 and 2018, they measured EMG potentials from different activities in over a thousand participants in diverse projects. Using these measurements, a reference point of maximal voluntary contraction was set. With the suit, we were able to measure values of muscle loading for both hands and compare them to the maximum voluntary contraction of the larger sample. A recommendation was made that when muscle loading exceeds 30%, additional breaks should be considered, and loading over 50% was not recommended (Myontec Ltd.)

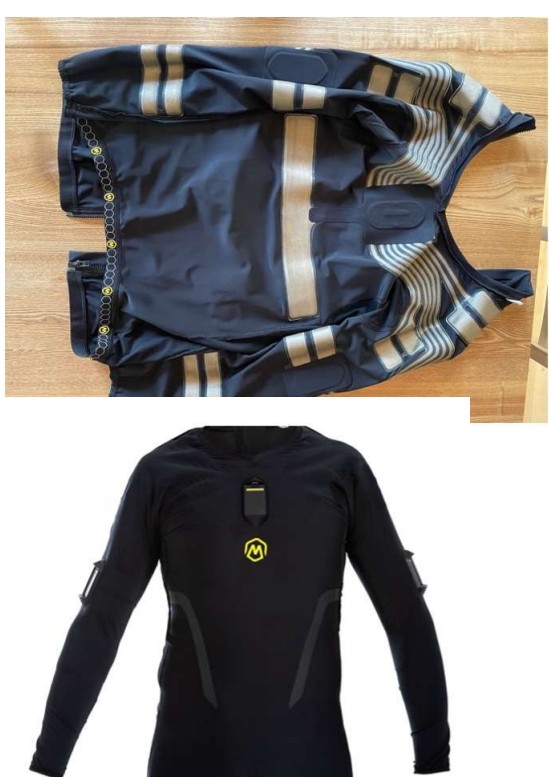

**Figure 8.** Ergosuit (Myontec Ltd., Kuopio, Finland).

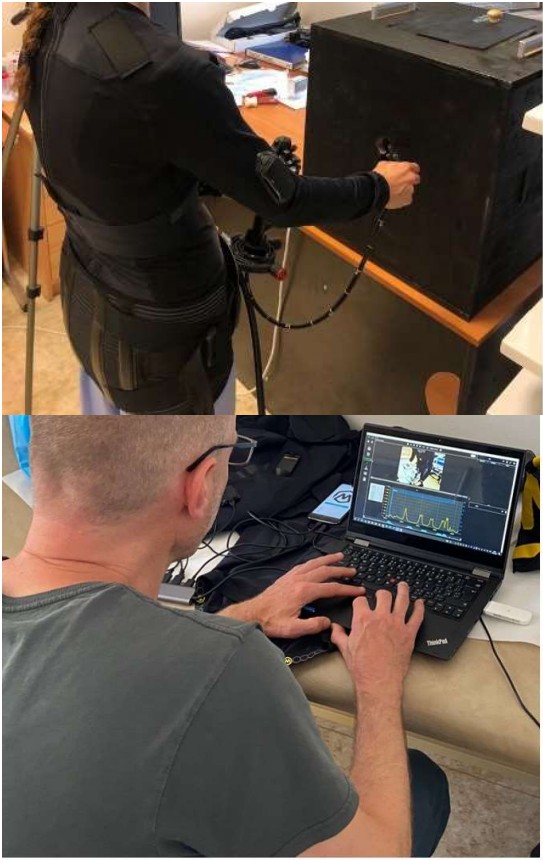

**Figure 9.** Experiments with the training box and data gathering.

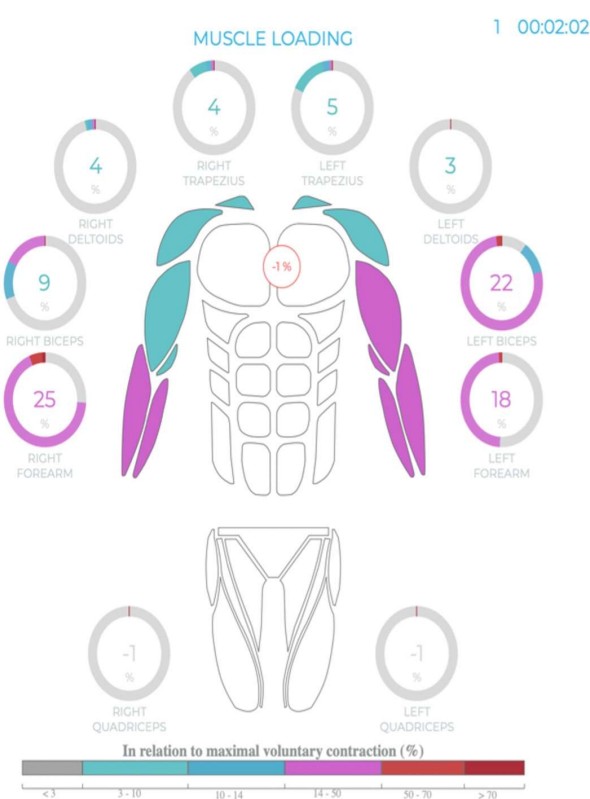

**Figure 10.** Diagram of muscle loading.

Figure 11 describes muscle tension and the level of microbreaks (Figure 11). The first two graphs describe muscle tension for the left and right arms, respectively. A colored scale shows the intervals of muscle tension for both arms, with the recommendation that continuous muscle tensity should not exceed 4 to 8 min. The second part of the diagram shows circular graphs with percentage values, indicating the percentage of microbreaks in the main muscle groups for both hands. The recommendation from Myontec Ltd. was that microbreaks should exceed 5% of the working phase, meaning that during the working phase, muscles should relax for more than 5% of the working period.

### 2.4. Muscle Tension and Microbreak Measurements with Support System

Lastly, we investigated if we could influence muscle loading, muscle tension, and microbreaks using a belt-like endoscopic holder, which is a commercially available support system for endoscopists. With the use of a belt-like holder, the endoscopist transfers the weight of the endoscope from the left hand to the body. It allows the hands to be more relaxed, as the scope position can be altered with body movements. ScopeDoc (COOK Medical, Bloomington, IN, USA) and Endojoystick (X.G.L.U s.r.o., Prague, Czech Republic) are commercially available support systems. Following careful examination of both systems, we decided to use Endojoystick, due to its extended maneuvering abilities. Endojoystick is a belt-like support system with joystick capabilities. The endoscope is mounted on the holder via the central piece. It offers a locked and a freehand position (Figure 12).

### 2.5. Participants

There were 16 healthcare workers included in this study. Participants were chosen and distributed according to their endoscopic skills: we included both experienced endoscopists (N = 8) and non-endoscopists (N = 8). It was our intention to make a heterogenous group including both sexes, with ages that ranged from 25 to 62 years. Both groups performed the same standardized tasks, firstly without the holder, and then, with the belt-like support system with joystick capabilities. Before testing, each participant received

formal training on how to handle the endoscope and of the tasks to be performed. The process of formal training, suit preparations, and application took approximately 1 h per participant. Guidance and assistance in the form of verbal instruction were provided before and during the experiments. Each participant was treated according to ethical standards. Before the clinical trials, the approval of the ethical committee was obtained.

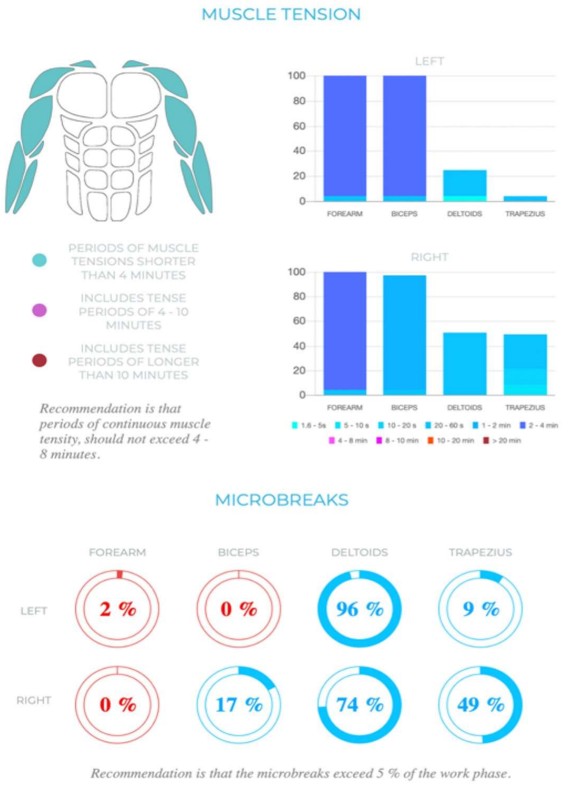

**Figure 11.** Diagram of muscle tension and microbreaks.

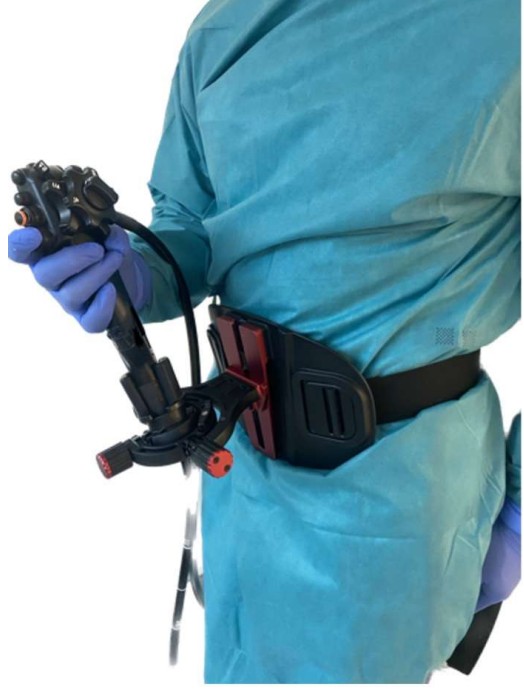

**Figure 12.** A person holding an endoscope with the Endojoystick belt-like endoscopic holder.

### 2.6. Statistics

For data processing and statistical analysis, R-Studio (v. 2022.07.2) software was used. We used the two-sided paired Student's *t*-test to compare observations between the groups. The *t*-test is a statistical hypothesis that takes samples from two groups to determine if there is a significant difference between the means of the two groups. It compares both the sample mean and standard deviations while considering sample size and the degree of variability in the data. However, there are several considerations to be made when using the t-test. Larger sample sizes give more accurate results. With smaller sample sizes, the test may have less statistical power, meaning it is less likely to detect a significant difference between the samples if one truly exists. Secondly, the *t*-test assumes that the data in each sample are normally distributed and have similar variances. We assumed that our observational group was normally distributed according to age. We therefore also assumed that their muscle ability measurements were normally distributed. We compared the means of the observed groups and set our hypothesis accordingly:

**H0:** *There is no significant difference between the means of the two groups.*

**H1:** *There is a significant difference between the means of the two groups.*

- If $p > 0.05$, it suggests that there is not enough evidence to conclude a significant difference between the two groups (FAIL).
- If $p < 0.05$, it indicates that the difference between the two groups is statistically significant (PASS).

## 3. Results

### 3.1. EMG Potentials during Endoscopy

The initial measurements of the EMG potentials during basic endoscopic maneuvers showed activation of the muscles of the left arm and shoulder/neck area. Activation was seen in the flexors, extensors, biceps, and the deltoid and trapezoid muscles. The analysis of the EMG curves and video recordings suggested higher muscle loading and fewer microbreaks in the biceps and trapezoid muscles. At this point no numerical value was obtained and only graphical and video analysis was available. Through these initial experiments, we were able to identify the muscles involved in endoscope handling and outline the potential muscle groups that were overloaded.

### 3.2. Differences between Endoscopists and Non-Endoscopists

After task standardization and upgrades to the measuring equipment, we measured the EMG potentials, muscle tension (loading), and number of microbreaks taken in endoscopists and non-endoscopists. A detailed report was created for each participant. In the endoscopist group, the mean muscle loading was 9.4% in the left forearm and 9.5% in the left biceps, in relation to the maximal voluntary contraction of a bigger sample. The maximal voluntary contraction was set by the Premedis company, as explained earlier in Figure 6. The mean muscle loading values in the left deltoid and left trapezius were 1.7% and 3.1%, respectively. The mean level of microbreaks in the left forearm was 31.9% of the working phase, and in the left biceps, 22.7%. The mean levels of microbreaks in the left deltoid and left trapezius were 96.6% and 32.6%, respectively. In the non-endoscopist group, the mean muscle loading in the left forearm was 16.5%, and in the left biceps, 11.3%, in relation to the maximal voluntary contraction of a bigger sample. The mean muscle loading values in the left deltoid and left trapezius were 3.9% and 4.3%, respectively. The mean level of microbreaks in the left forearm was 18.6%, and in the left biceps, 3.1%. The mean levels of microbreaks in the left deltoid and left trapezius were 79.3% and 60.1%, respectively. The data are presented in Tables 1 and 2.

**Table 1.** Endoscopists' mean muscle tension and microbreaks.

| Endoscopists | |
| --- | --- |
| Mean muscle tension % | **Freehand** |
| Left forearm | 9.4 |
| Left biceps | 9.5 |
| Left deltoid | 1.7 |
| Left trapezius | 3.1 |
| Mean microbreaks % | **Freehand** |
| Left forearm | 31.9 |
| Left biceps | 22.7 |
| Left deltoid | 96.6 |
| Left trapezius | 32.6 |

**Table 2.** Non-endoscopists' mean muscle tension and microbreaks.

| Non-endoscopists | |
| --- | --- |
| Mean muscle tension % | **Freehand** |
| Left forearm | 16.5 |
| Left biceps | 11.3 |
| Left deltoid | 3.9 |
| Left trapezius | 4.3 |
| Mean microbreaks % | **Freehand** |
| Left forearm | 18.6 |
| Left biceps | 3.1 |
| Left deltoid | 79.3 |
| Left trapezius | 60.1 |

### 3.3. Differences between Groups When Using the Belt-like Support System

In the second part of the experiment, we introduced the belt-like support system. The same set of tasks were performed by all participants. We measured the EMG potentials, muscle tension (loading), and levels of microbreaks in endoscopists and non-endoscopists. In the endoscopist group, the mean muscle loading in the left forearm was 8.6%, and in the left biceps, 6.4%, in relation to the maximal voluntary contraction of a bigger sample. The mean muscle loading values in the left deltoid and left trapezius were 1.4% and 1.4%, respectively. The mean level of microbreaks in the left forearm was 48.1%, and in the left biceps, 40.4%. The mean levels of microbreaks in the left deltoid and left trapezius were 99.1% and 94.9%, respectively. In the non-endoscopist group, the mean muscle loading in the left forearm was 13.3%, and in the left biceps, 8.3%, in relation to the maximal voluntary contraction of a bigger sample. The mean muscle loading values in the left deltoid and left trapezius were 5.4% and 2.1%, respectively. The mean level of microbreaks in the left forearm was 34.4%, and in the left biceps, 32.5%. The mean levels of microbreaks in the left deltoid and left trapezius were 53.9% and 85.6%, respectively. The data are presented in Tables 3 and 4.

**Table 3.** Endoscopists' mean muscle tension and microbreaks with belt support holder.

| Endoscopists | |
| --- | --- |
| Mean muscle tension % | **Belt holder** |
| Left forearm | 8.6 |
| Left biceps | 6.4 |
| Left deltoid | 1.4 |
| Left trapezius | 1.4 |
| Mean microbreaks % | **Belt holder** |
| Left forearm | 48.1 |
| Left biceps | 40.4 |
| Left deltoid | 99.1 |
| Left trapezius | 94.9 |

**Table 4.** Non-endoscopists' mean tension and microbreaks with belt support holder.

| Non-endoscopists | |
|---|---|
| **Mean muscle tension %** | **Belt holder** |
| Left forearm | 13.3 |
| Left biceps | 8.3 |
| Left deltoid | 5.4 |
| Left trapezius | 2.1 |
| **Mean microbreaks %** | **Belt holder** |
| Left forearm | 34.4 |
| Left biceps | 32.5 |
| Left deltoid | 53.9 |
| Left trapezius | 85.6 |

*3.4. Statistical Analysis*

We compared the mean muscle tension and microbreaks in all participants with and without the belt-like support system; endoscopists with and without the belt-like support system; and non-endoscopists with and without the belt-like support system. Due to the small sample size, the results were inconclusive when comparing endoscopists and non-endoscopists with and without the belt-like support system. There was a statistically significant difference when comparing all participants with and without the belt-like support system in muscular tension and in the level of microbreaks in the left forearm, left biceps, and left trapezius. There was no statistically significant difference in muscle tension and in the level of microbreaks in the deltoid muscle. As stated before, if there was not enough evidence to conclude a significant difference between the two groups and $p > 0.05$, we assigned the notation FAIL. If the evidence was sufficient and $p < 0.05$, we assigned the notation PASS. The *p*-values in numerical and FAIL/PASS form are presented in Table 5.

**Table 5.** *p*-values in numerical and FAIL/PASS form for all participants, endoscopists, and non-endoscopists.

| | *p*-Values | | |
|---|---|---|---|
| | **All** **(with vs. without Belt)** | **Endoscopists** **(with vs. without Belt)** | **Non-Endoscopists** **(with vs. without Belt)** |
| LF m. tension | 0.012260 | 0.248100 | 0.026560 |
| LB m. tension | 0.000790 | 0.041320 | 0.011690 |
| LD m. tension | 0.278200 | 0.172300 | 0.190600 |
| LT m. tension | 0.003163 | 0.009604 | 0.076940 |
| LF microbreaks | 0.028460 | 0.106400 | 0.169400 |
| LB microbreaks | 0.006564 | 0.175600 | 0.022870 |
| LD microbreaks | 0.105500 | 0.172300 | 0.066260 |
| LT microbreaks | 0.000506 | 0.006452 | 0.020850 |
| | All | Endoscopists | Non-Endoscopists |
| LF m. tension | PASS | FAIL | PASS |
| LB m. tension | PASS | PASS | PASS |
| LD m. tension | FAIL | FAIL | FAIL |
| LT m. tension | PASS | PASS | FAIL |
| LF microbreaks | PASS | FAIL | FAIL |
| LB microbreaks | PASS | FAIL | PASS |
| LD microbreaks | FAIL | FAIL | FAIL |
| LT microbreaks | PASS | PASS | PASS |

Figure legend for Table 5 and Figures 13 and 14: LF = left forearm; LB = left biceps; LD = left deltoideus; LT = left triceps.

- Left forearm muscle tension

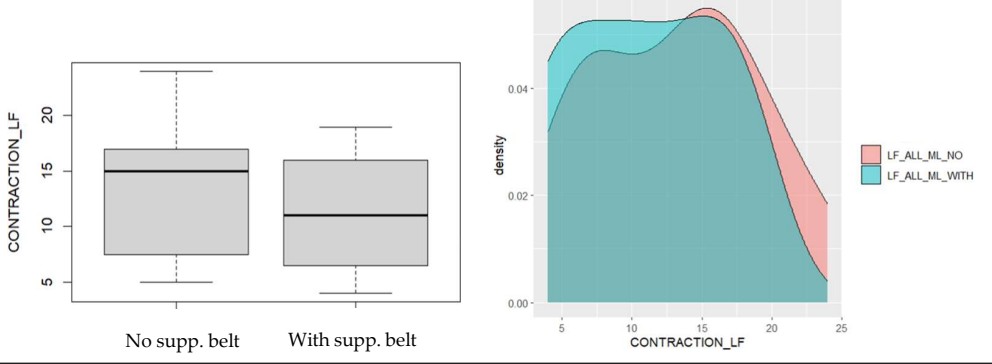

- Left biceps muscle tension

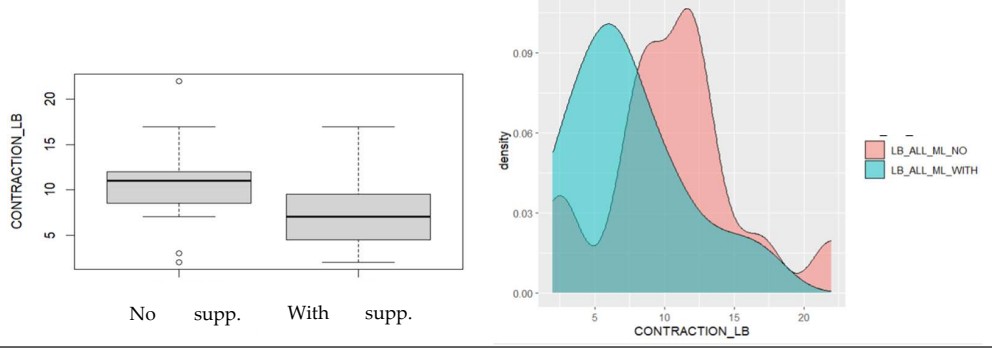

- Left deltoid muscle tension

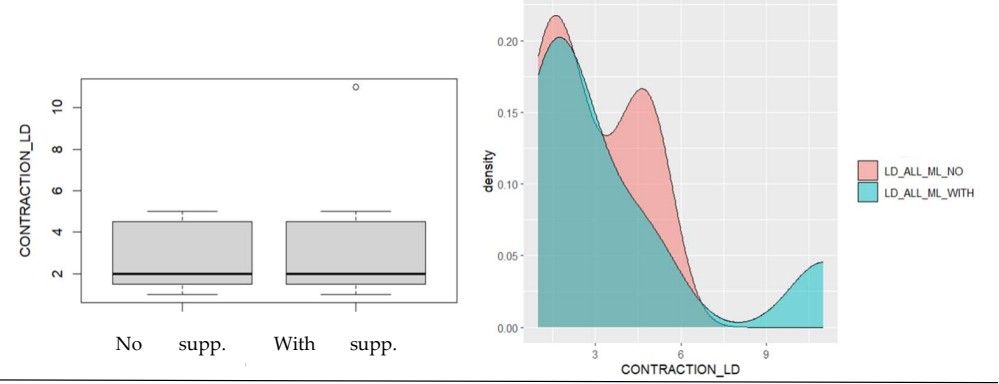

- Left trapezius muscle tension

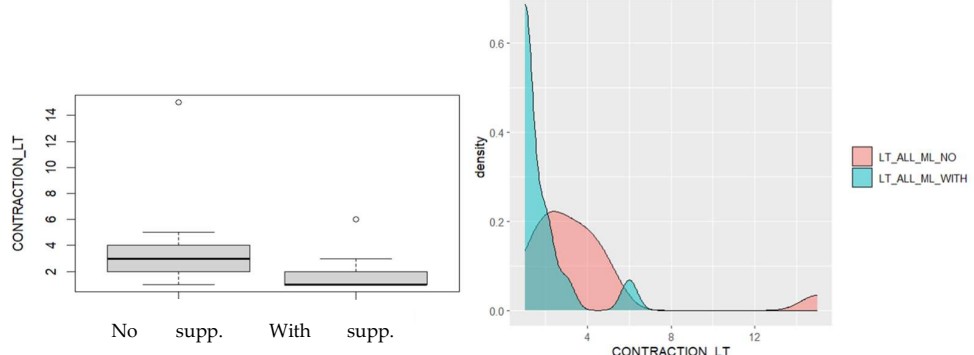

**Figure 13.** Histograms and density graphs for left upper-extremity muscle tension.

- Left forearm level of microbreaks

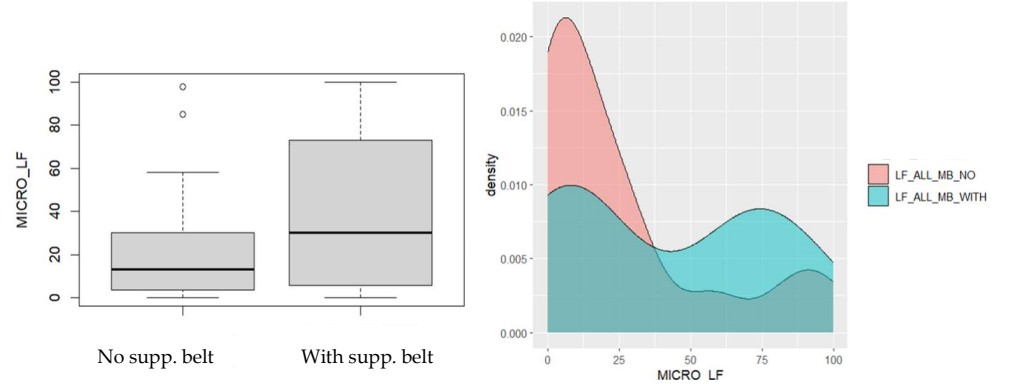

- Left biceps level of microbreaks

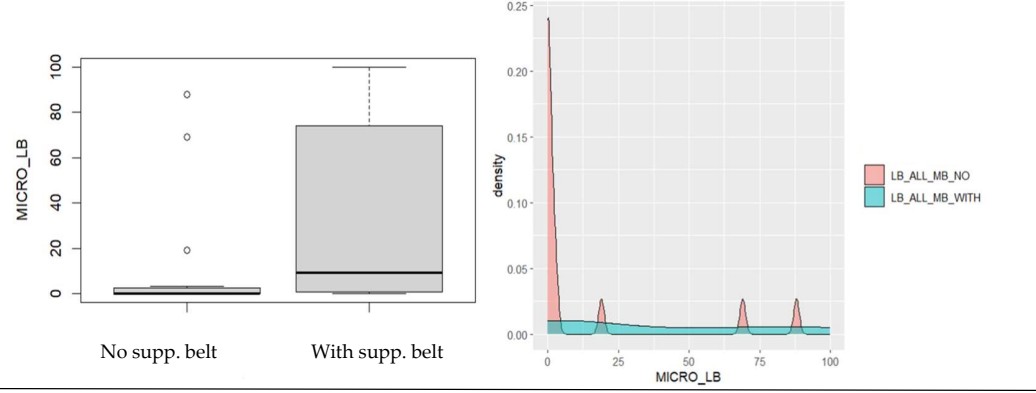

- Left deltoid level of microbreaks

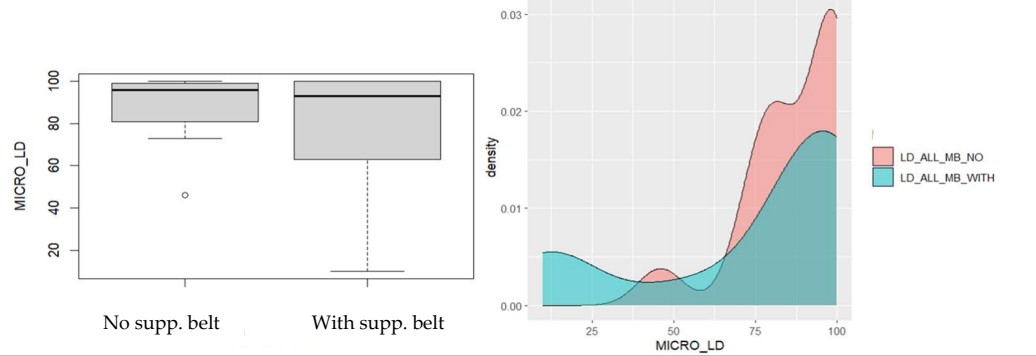

- Left trapezoid level of microbreaks

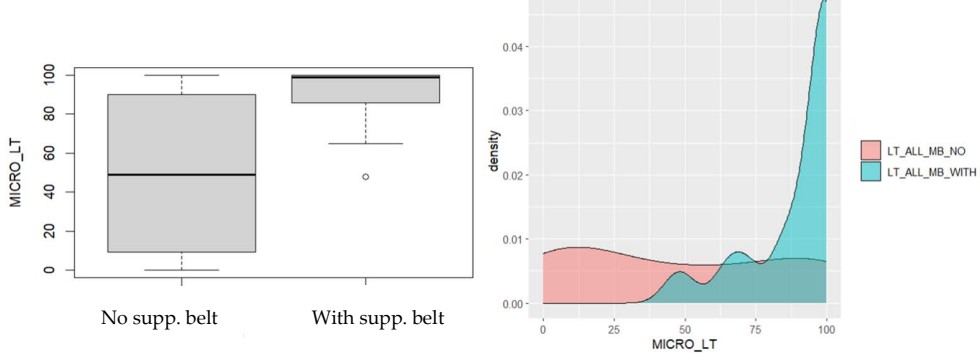

**Figure 14.** Histograms and density graphs for left upper-extremity microbreaks.

Figures 13 and 14 show a comparison of all participants without and with the belt support system in the form of histograms and density graphs for each muscle group.

## 4. Discussion

A basic set of recommendations and ergonomic principles has been provided by multiple authors and the ASGE (American Society for Gastrointestinal Endoscopy). All recommendations follow a similar structure, including guidance on preparing the endoscopy room, approaching endoscopic procedures, and implementing planning strategies. The common feature of all these recommendations is the focus on how to minimize the risk of MSK injuries during endoscopic procedures, missing the implementation of the possibilities of additional accessories or other technical approaches regarding injury prevention. Studies have shown that despite careful adherence to these guidelines, injuries still happen [3,6,7]. Solutions to prevent or eliminate occupational injuries in this area are required.

In the first part of our study, we analyzed an endoscopist performing basic endoscopic movements. Through careful examination and cooperation with a physiotherapist, we outlined the muscles involved during endoscopic procedures. With surface EMG sensors and video recordings, we confirmed the muscles working in these areas. Due to the complex design with multiple EMG skin sensors, video synchronization, the small number of participants, and the absence of a standardized movement, our data and findings were inconclusive.

We then standardized the movement of an endoscopist for further experiments with the design of the endoscopic task box. EMG data gathering was performed to a higher level by using a smart suit with video synchronization. The inclusion of a greater number of participants allowed more data to be compared. Also, an upgrade in software and access to a larger data base (Premedis) enabled a better understanding of our gathered data. Muscle overloading and a reduced level of microbreaks were present in the left forearm, left biceps, and left trapezius in both groups. The group of endoscopists presented a very comparable set of data, with very low diversity. The non-endoscopist group also achieved very similar data, but with larger diversity, presenting a more variable and dynamic set of data. Our explanation of this data divergence is that non-endoscopists lacked the experience of performing the procedures and so had no muscle memory when performing endoscopy. Hence, the level of stress and muscle tone were higher and of a different nature in each participant, resulting in variable muscle load and microbreaks.

After the introduction of the belt-like endoscopic holder, differences were observed in both groups. We tested for statistically significant differences in muscle tension and levels of microbreaks inside both groups. Due to the small group sample size, there was no significant difference. We increased the observed sample by comparing all participants without and with the belt support system, regardless of their endoscopic background. A statistically significant difference ($p < 0.05$) in muscle tension and in the levels of microbreaks was present in the muscles of the left forearm, biceps, and trapezius muscle. No statistically significant difference was observed in muscle tension and the level of microbreaks in the left deltoid ($p > 0.05$). As this is a pilot study, our main limitation was the small number of participants and lack of statistical power within the groups. We demonstrated statistically significant differences when comparing all participants; however, a study with a higher number of participants is needed to verify the obtained result.

Muscles can be roughly divided into active and passive muscles. Active muscles make our bodies move, whereas passive muscles participate in equilibrium and body stabilization. The main complaints of endoscopists are pain in the hands, arms, and neck area. When moving, active muscles are depleted of energy before the passive ones are. After an undefined period exercising a specific set of moves, the passive muscles also become energetically depleted and overloaded, and shift to anaerobic metabolism. This can result in chronic changes such as fibrosis, atrophy, and shortening of the muscles themselves. Adjacent structures can be impinged, resulting in pain and movement disability. When applying this principle to an endoscopist, stabilizers of the neck play an important role.

The main stabilization muscles in this area are the scalene, sternocleidomastoids, levator scapulae, and trapezius. As the whole brachial plexus runs adjacent to and through these muscles, the overloading of these muscles can cause issues with hand, arm, and neck pain, and movement disability. With the surface EMG method, we were able to record the activity of the active muscles. However, neck stabilizers are in the deeper layers and a more invasive approach would be needed (needle EMG) to define their activity. However, such invasive measures risk damage to the muscles and adjacent structures, and cause pain to the participant. This was therefore an inappropriate choice for our study.

After careful examination of the data, it is our understanding that increased levels of muscle loading and decreased levels of microbreaks in the deltoid imply that this muscle has a different role to the others, and that the muscle participates only in shoulder motion [15]. The deltoid muscle does not play an active role in body posture and body stabilization. We were therefore unable to demonstrate a statistically significant difference in muscle tension and the levels of microbreaks of the deltoid muscle with and without the use of the support belt.

## 5. Conclusions

The prevention of MSK injuries in endoscopists during endoscopic procedures is a challenge. As the root of all problems lies within the design and single size of endoscopic devices, the most logical solution is to alter the approach to endoscopy and the endoscopic equipment. However, progress is slow and limited by organizational and financial reasons.

Through our study, we were able to show that the introduction of an ergonomic accessory may help to prevent MSK injuries in endoscopists during endoscopic procedures. With the use of a belt-like support system, we were able to demonstrate a statistically significant difference in the level of muscle tension and microbreaks in affected muscle areas. Hence, it is our belief that use of a belt-like support system may play an important role in occupational injury prevention amongst GI endoscopists.

**Author Contributions:** Conceptualization, T.D. and J.H.; methodology, T.D.; investigation, T.D.; writing—original draft preparation, T.D.; writing—review and editing, I.C. and J.H.; supervision, J.H.; project administration, I.C.; funding acquisition, T.D. and J.H. All authors have read and agreed to the published version of the manuscript.

**Funding:** This research was funded by the Charles University Grant Agency (GAUK), grant number 412622.

**Institutional Review Board Statement:** This study was conducted in accordance with the Declaration of Helsinki and approved by the Institutional Review Board (or Ethics Committee) of Eticka komise Fakulni Nemocnice Kralovski Vinohrady (EK-VP/60/0/2021 on 3 November 2021) for studies involving humans.

**Informed Consent Statement:** Informed consent was obtained from all subjects involved in this study.

**Data Availability Statement:** Data available on request due to restrictions eg privacy or ethical. The data presented in this study are available on request from the corresponding author. The data are not publicly available due to privacy reasons.

**Conflicts of Interest:** The authors declare no conflicts of interest.

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
