# Peer review of "Prevention of Musculoskeletal Injuries in Gastrointestinal Endoscopists"

_gastroent, doi:10.3390/gastroent15020021_

Round 1
Reviewer 1 Report
Comments and Suggestions for Authors
This study addresses a critical gap in the literature by investigating the impact of endoscopy procedures on musculoskeletal stress and the potential benefits of support systems. Gastroenterologists face increasing risks of musculoskeletal injuries, yet there is a scarcity of research considering their burden. The authors' approach of identifying stressed muscle groups, measuring activity through electromyography, and assessing the effectiveness of support systems provides valuable insights.
Despite the small sample size, the study yields significant findings, particularly regarding the use of a belt-like support system. The comparison between participants with and without the support system demonstrates statistically significant differences in muscle tension and microbreaks in several muscle groups, emphasizing the potential benefits of such ergonomic interventions.
The authors' hypothesis regarding the differential impact on the deltoid muscle and its relation to shoulder movements adds depth to the discussion, highlighting the complexity of musculoskeletal dynamics during endoscopy procedures.
Overall, this study significantly contributes to our understanding of musculoskeletal strain in gastrointestinal endoscopy and underscores the importance of ergonomic interventions in mitigating related injuries. It is recommended for acceptance, acknowledging its substantial utility in addressing the overlooked burden on endoscopists.
Reviewer 2 Report
Comments and Suggestions for Authors
In this study, authors aimed to identify the most stressed muscle groups during endoscopy using surface electromyographical (EMG) sensors and identified the number of microbreaks taken in specific muscles. Furthermore, they measured differences in the loading of these muscle groups with and without the use of special support systems such as a belt-like holder. This is a very innovative and important proposal as the number of long-term procedures is increasing with advances in therapeutic endoscopy. I have no criticisms on methodology, statistical analysis, and interpretation.
Reviewer 3 Report
Comments and Suggestions for Authors
Dear authors, Thank you so much for submitting your work. The article provide promising data on prevention of musculoskeletal injuries in GI endoscopists. However, there are some issues as below...
Major comments
1. Overall: The writing often lacks clarity and sharpness, and several sections are poorly organized or do not flow with the rest of the paper (e.g., Materials and Methods, for p-value, please just say p<0.05 is significant. etc.)
2. Methods: Line 295- Study design is not well described in the method section. N=8/group?. What age? sex? of participants. How did authors distribute participants, randomized or not? Please clarify it.
3. Results: Line420- Statistical analysis: Why do the authors show the stats separately? I am puzzled as to why the authors showed Pass/Fail in Table 5. Please show the stats together with each result (Table 1-4). It would be more readable.
4. Figure: Fig1 looks same as Fig 13. Please delete one of them.
5. Legend: Please don’t marge legends: “Figure 2 and 3”, Figure 5 and 6” should be separated. Please explain abbreviations in the figure legend (e.g. LF, LB, LD etc. in Table 5, Figure14)
Minor comments
1. Abbreviations must be defined upon first appearance. Please check through all the text, tables and figures. i.e...
a. Line 02: GI
b. Line 50: MSK
c. Line 78: EMG
2. Abbreviation needs address (i.e. city name, postal code, country name etc.).
3. There are some errors in grammars and format. Please check through all the text, tables and figures. i.e...
a. Extra spaces (looks like double-space): Line 57, 86,138,228
b. Keywords don’t need #: Line 31: “GI endoscopy 1” ==> GI endoscopy
c. Table 1 ==> Table2, Table2==> Table1
4. Line 310: What version for R-Studio?
Comments on the Quality of English LanguageQuality of English should be fine, but the writing lacks clarity.
Round 2
Reviewer 3 Report
Comments and Suggestions for Authors
Dear Authors, I appreciate your all modifications. Almost all issues which I pointed out have been corrected. I still see some errors. Please collect them.
· Title: gastrointestinal (GI) endoscopists: I don’t think you need (GI). Please consider to say gastrointestinal endoscopists.
· Please separate Fig2 and Fig3 and show in order (i.e. Fig2 with its legend, Fig3 with its legend).
· Please separate Fig5 and Fig6 and show in order (i.e. Fig5 with its legend, Fig6 with its legend).
Author Response
Dear reviewer, we corrected all suggestions, thank you for your input.